# Repression of TERRA Expression by Subtelomeric DNA Methylation Is Dependent on NRF1 Binding

**DOI:** 10.3390/ijms20112791

**Published:** 2019-06-07

**Authors:** Gabriel Le Berre, Virginie Hossard, Jean-Francois Riou, Anne-Laure Guieysse-Peugeot

**Affiliations:** Structure et Instabilité des Génomes, Muséum National d’Histoire Naturelle, CNRS UMR 7196, INSERM U1154, 43 rue Cuvier, F-75005 Paris, France; gabriel.le-berre@mnhn.fr (G.L.B.); virginie.hossard@mnhn.fr (V.H.); jean-francois.riou@mnhn.fr (J.-F.R.)

**Keywords:** telomeres, TERRA, DNA methylation, NRF1, targeted demethylation, epigenetic regulation

## Abstract

Chromosome ends are transcribed into long noncoding telomeric repeat-containing RNA (TERRA) from subtelomeric promoters. A class of TERRA promoters are associated with CpG islands embedded in repetitive DNA tracts. Cytosines in these subtelomeric CpG islands are frequently methylated in telomerase-positive cancer cells, and demethylation induced by depletion of DNA methyltransferases is associated with increased TERRA levels. However, the direct evidence and the underlying mechanism regulating TERRA expression through subtelomeric CpG islands methylation are still to establish. To analyze TERRA regulation by subtelomeric DNA methylation in human cell line (HeLa), we used an epigenetic engineering tool based on CRISPR-dCas9 (clustered regularly interspaced short palindromic repeats – dead CRISPR associated protein 9) associated with TET1 (ten-eleven 1 hydroxylase) to specifically demethylate subtelomeric CpG islands. This targeted demethylation caused an up-regulation of TERRA, and the enhanced TERRA production depended on the methyl-sensitive transcription factor NRF1 (nuclear respiratory factor 1). Since AMPK (AMP-activated protein kinase) is a well-known activator of NRF1, we treated cells with an AMPK inhibitor (compound C). Surprisingly, compound C treatment increased TERRA levels but did not inhibit AMPK activity in these experimental conditions. Altogether, our results provide new insight in the fine-tuning of TERRA at specific subtelomeric promoters and could allow identifying new regulators of TERRA.

## 1. Introduction

Telomeres, the nucleoprotein complexes located at the ends of chromosomes in eukaryotes, are formed by TTAGGG repeats associated with a complex of six proteins, the shelterin complex [1]. Telomeres prevent chromosome ends from being recognized as double-strand breaks in the DNA. In most human cells, telomeres shorten at each cell division, and when they become critically short, cells enter into cellular senescence resulting in growth arrest [2].

Subtelomeres are sequences enriched in repetitive DNA that is adjacent to telomeres [3]. Although telomeric and subtelomeric chromatin have heterochromatic features, these regions are transcribed into long noncoding telomere repeat containing RNAs (TERRA) [4]. TERRA transcripts are involved in several functions, notably heterochromatinization of subtelomeres [5,6], damage protection [7], and telomere length maintenance [8,9]. TERRA transcripts are heterogeneous in length (between 100 bases and 10 kilobases) and are transcribed from subtelomeric transcription start sites toward telomeres. In human cells, TERRA promoters on around twenty subtelomeres (notably Xq, Yq, 15q, 16p, and 9p) are highly CpG-rich and contain three conserved repetitive DNA tracts comprised of tandem repeat units of 61 base pairs (bp), 29 bp, and 37 bp (the so-called 61-29-37 repeats) [10]. These subtelomeric CpG islands are highly methylated in cancer telomerase-positive cell lines, and previous studies have shown that the knockout of genes encoding DNA methyltransferases DNMT1 and DNMT3b in the human HCT116 cell line leads to a demethylation of this sequence and to an increase in TERRA levels [10,11]. Moreover, hypomethylated subtelomeres and abnormally elevated TERRA levels are observed in cells isolated from patients affected by type I immunodeficiency, centromeric instability, and facial anomalies syndrome (ICF1), which is caused by mutations in the *DNMT3B* gene [12].

These results suggest that methylation of these subtelomeric CpG islands downregulates TERRA expression. A caveat is that in these studies, the DNA methyltransferases were inactivated, leading to a global hypomethylation of the genome associated with genome-wide transcriptional changes [13,14]. The increase of TERRA levels could be due, at least partially, to indirect effects such as impaired degradation of these transcripts. Thus, to address the epigenetic regulation of TERRA expression, it is necessary to modify the methylation status specifically at these subtelomeric CpG islands. These investigations are all the more relevant as several reports have recently challenged common views on the regulation of gene expression by DNA methylation. The expression of some genes has been shown to be uncorrelated with the methylation status of their upstream CpG islands [15]. For example, in acute myeloid leukemia derived cell lines, *AWT1* (a transcript from an alternative promoter of the *WT1* (Wilms tumor 1) gene) is highly expressed despite the hypermethylation of its promoter [16]. Even when an inverse correlation is observed between DNA methylation and gene expression across different cell lines, DNA methylation is not necessarily the cause of the gene repression. This was shown in breast cancer cells where aberrant promoter hypermethylation generally occurs in genes already repressed in the tissue of origin and is, therefore, not responsible for their repression [17].

Different mechanisms by which methylation of upstream CpG islands influences gene regulation have been characterized [18], and one of the well-described mechanisms is the alteration of the transcription factor binding by DNA methylation. For example, in vivo and in vitro studies have shown that cytosine methylation of the recognition site of the transcription factor NRF1 (nuclear respiratory factor 1) inhibits its binding [19,20,21]. Interestingly, NRF1 was also shown to bind to the subtelomeric CpG islands and is a positive regulator of TERRA expression [22]. Sequence prediction indicates that each 29 bp repeat in TERRA promoter regions contains two putative NRF1 binding sites. One of these recognition sites is also located in the murine *ASZ1* (ankyrin repeat, SAM and basic leucine zipper domain containing 1) promoter, and bandshift experiments showed that affinity of NRF1 for this site decreases when cytosines are methylated [23]. NRF1 was originally described to be involved in regulation of mitochondrial biogenesis and oxidative phosphorylation, and its activity increases when the AMPK pathway is activated [24]. Despite the link between the AMPK/NRF1 pathway and TERRA transcription [22], the influence of the DNA methylation on the NRF1-dependent TERRA regulation has not yet been determined.

In this study, we used CRISPR/dCas9 (clustered regularly interspaced short palindromic repeats – dead CRISPR associated protein 9) to target a DNA demethylase to subtelomeric CpG islands in the human cell line (HeLa) to analyze the epigenetic regulation of TERRA expression. We used the SunTag system (called the CRISPR-dCas-TET1 system hereafter) developed by Morita et al. [25] to recruit the exogenous ten-eleven 1 hydroxylase TET1 to the locus targeted by the RNA guide oligonucleotides. TET1 catalyzes the oxidation of methylated cytosine to 5-hydroxymethylcytosine, leading to its demethylation [26]. We found that the targeted demethylation of subtelomeric CpG islands resulted in an NRF1-dependent increase in TERRA expression. In line with this, global genomic DNA demethylation induced by inhibition of DNA methyltransferases after 5-aza-deoxycytidine (5-aza-dC) treatment was associated with an increase of the NRF1 binding at the subtelomeric CpG islands. Moreover, we found that the treatment of HeLa and HCT116 cells with compound C, commonly used as an inhibitor of the AMPK pathway, induced an increase of the TERRA level that we discovered was not due to inhibition of AMPK.

## 2. Results

### 2.1. Development of A Method to Analyze The Methylation of The 29 bp Repeats

The TERRA promoters located on chromosomes Xq, Yq, 16p, 15q, and 9p were chosen as a model for the family of subtelomeres containing 61-29-37 repeats embedded in a CpG island (Figure 1A). To determine if TERRA expression depends on the methylation of the subtelomeric CpG islands, we used a CRISPR-dCas-TET1 system to induce targeted demethylation.

Previous promoter reporter assays have shown that the activity of a TERRA promoter containing the 61-29-37 repeats is mainly dependent on the presence of 29 bp repeat [10,27], suggesting that these repeats contain cis-regulator elements. Thus, epigenetic regulation of TERRA could depend, at least in part, of the methylation of 29 bp repeats. It was therefore necessary to have a method to evaluate the methylation level of the 29 bp repeat and to determine if the CRISPR-dCas9-TET1 system used in this study efficiently demethylated this region.

Two major techniques have been previously used to study methylation of this subtelomeric CpG islands. The first, southern-blot analysis of a cleaved fragment after digestion with methyl-sensitive restriction enzymes, is not accurate enough to specifically evaluate the methylation of the 29 bp repeat regions [10]. The second technique is bisulfite conversion of unmethylated cytosines in thymines followed by PCR amplification of the locus of interest. Given that TERRA promoters are highly repetitive, CpG-rich, and only partially conserved on different subtelomeres, this method has been applied to study the methylation status of only a few CpG of this region not located in 29 bp repeats [5]. Despite many attempts under different conditions, we have not succeeded in amplifying the 29 bp repeat regions after bisulfite conversion (data not shown).

Because these methods are not appropriate for studying the methylation of the tandem repeat region, we have developed a new method that does not require bisulfite conversion and is more sensible than southern-blot analysis. Our approach was inspired by the methylation-sensitive restriction enzyme PCR (MSRE-PCR) method, which consists of DNA digestion with a methylation sensitive endonuclease, followed by a PCR amplification with primers complementary to regions on either side of the recognition site of the endonuclease [28]. Thus, the sequence of interest is amplified only when the site is methylated. The expected effect of the CRISPR-dCas9-TET1 system was a demethylation of the subtelomeric CpG islands, which are basically hypermethylated in human telomerase-positive cell lines [10]. To have a positive signal and detect a PCR product in the presence of unmethylated CpG, the MSRE-PCR method was modified, so that PCR amplification would occur only if the recognition site of the endonuclease HpaII, present in each of the 29 bp repeats, is demethylated (Figure 1B). To achieve this, a specific elongation of one of the strands was carried out after the cleavage by HpaII using a template oligonucleotide (OT). After this elongation step, a PCR amplification was carried out with a primer (PF) complementary to sequence added to the 3′ side of the cleavage site and a primer (PR) complementary to a region immediately upstream of the 29 bp repeat. The appearance of PCR products indicates the presence of demethylated HpaII recognition site in 29 bp repeats. Different sizes of PCR products are expected, depending on the position of the demethylated recognition sites of HpaII. The size of the shortest PCR product is 55 bp, and the sizes of the longer PCR products are 55 bp and a multiple of 29 bp. A PCR band of 178 bp can also be detected, due to a duplication of the sequence complementary to the primer PR in the subtelomere 1p.

As a proof of principle, we applied this method to study subtelomeric DNA methylation in HeLa (human cervix carcinoma) cells treated or not with the DNA methyltransferase inhibitor 5-aza-dC for 72 h. The expected profiles of PCR bands appeared from DNA extracted from cells treated with 5-aza-dC and digested with HpaII (Figure 1C, lane 1). When DNA was digested with MspI, which is an isoschizomer of HpaII insensitive to methylation (lane 5), the PCR product of 55 bp was the most intense band, but longer PCR products also appeared, suggesting that the digestion with MspI was incomplete. The expected PCR products did not appear in samples not treated with restriction enzyme (lanes 2, 4, and 6), proving that this method does not generate false positive results. Very weak PCR bands were detected when DNA extracted from untreated HeLa cells was digested with HpaII (lane 3), showing that the 29 bp repeats are highly methylated in this cell line.

### 2.2. Telomerase-Positive and alternative lengthening of telomeres (ALT) Cells Are Differently Methylated at The Subtelomeric 29 bp Repeat

Two telomere length maintenance mechanisms have been described in cancer cells: enhanced telomerase activity and a recombination-based alternative lengthening of telomeres (ALT). Previous reports have shown that there is a lower level of methylation of different subtelomeric loci in ALT cell lines than in telomerase-positive cell lines [29,30]. To compare the methylation status of the 29 bp repeats in telomerase-positive and in ALT cell lines, we used our method to evaluate methylation at the 29 bp repeats in three telomerase-positive cell lines (HeLa, T98G, and HCT116) and two ALT cell lines (U2OS and SaOS-2). To perform a semi-quantitative analysis, the PCR was stopped during the linear phase of amplification. The intensity of the PCR bands obtained when DNA was digested by HpaII was higher in the two ALT cell lines than in the three telomerase-positive cell lines (Appendix A). This result indicates that the 29 bp repeats are less methylated in ALT cell lines than in telomerase-positive cell lines.

### 2.3. Specific Demethylation of The Subtelomeric CpG Islands Is Induced by The CRISPR-dCas9-TET1 System

To specifically demethylate the subtelomeric CpG islands in HeLa cells, we used a SunTag system developed by Morita et al. [25] that can recruit multiple TET1 catalytic domains on the target sites (called, in our study, the CRISPR-dCas9-TET1 system). To determine whether the CRISPR-dCas9-TET1 system specifically demethylates the 29 bp repeats, HeLa cells were transfected with the vectors coding the proteins of this SunTag system and with vectors for expression of the guide RNAs (gRNAs). Three gRNAs targeting three different sites on the subtelomeric CpG islands were expressed together in the cells (Figure 1A, sub1, sub2, and sub3). After transfection, methylation of the 29 bp repeats was assessed using our method (Figure 1B). We performed 35 PCR cycles in order to ensure detection of any demethylated sequence. The amount of PCR product corresponding to demethylated 29 bp repeats was higher in the presence of active TET1 and gRNAs targeting subtelomeres compared to mock transfection controls (Figure 2A), indicating that this tool enabled to demethylate the 29 bp repeats. Expression of a gRNA with a sequence not found in the human genome was used to control a potential global demethylation induced by the system. The CRISPR-dCas9-TET1 system did not induce a global demethylation of the genome, as shown by the absence of demethylation of the 29 bp repeats with the control gRNA (Figure 2A). Moreover, transfection with subtelomeric-targeted gRNAs and vectors in which the coding sequence for TET1 was mutated (H1671Y and D1673A) to produce catalytically dead enzyme did not induce demethylation of the 29 bp repeat, indicating that demethylation of the 29 bp repeats induced by the CRISPR-dCas9-TET1 system is dependent on the catalytic activity of the demethylase. 

To compare the efficiency of the specific demethylation induced by our epigenetic engineering tool with the demethylation induced by 5-aza-dC treatment, DNA from cells treated under these conditions was evaluated with PCR stopped during the linear phase of amplification (30 cycles). The amount of PCR product corresponding to demethylated 29 bp repeats was significantly higher when HeLa cells were treated with 5-aza-dC than when they were transfected with the CRISPR-dCas9-TET1 vectors and subtelomeric-targeted gRNAs (Appendix A). Thus, the demethylation of the 29 bp repeats induced by the CRISPR-dCas9-TET1 system was much weaker than that induced by 5-aza-dC treatment. Noteworthy, the signal obtained from DNA digested with the methyl-insensitive restriction endonuclease MspI was higher than that obtained from DNA from 5-aza-dC-treated cells and digested with HpaII, suggesting that the demethylation of the 29 bp repeats induced by 5-aza-dC treatment was not complete (Appendix A). 

### 2.4. Targeted Demethylation of The Subtelomeric CpG Islands Increases TERRA Expression

To determine whether the specific demethylation of the subtelomeric CpG islands induced by the CRISPR-dCas9-TET1 system affected TERRA expression from the subtelomeres targeted by the RNA guides, the level of TERRA transcribed from subtelomeres Xq, Yq, 15q, and 9p was evaluated by RT-qPCR using the primers, TF and TR. The active TET1 associated with sub gRNAs induced a 5.6-fold increase in TERRA level, compared to mock transfection (Figure 2B, upper panel). On the other hand, no change of TERRA level was observed with catalytically dead TET1 or control gRNA. To determine potential bystander effects of our epigenetic engineering tool on TERRA expression from subtelomeres not targeted by the gRNAs used, we analyzed TERRA levels from Xp and Yp subtelomeres, which are transcribed [4] but do not contain sequence complementary to the RNA guides. Our targeted demethylation system did not have a significant effect on levels of TERRA transcripts produced from Xp or Yp (Figure 2B, lower panel). These results demonstrate that specific demethylation of the subtelomeric CpG islands induces an increase of the TERRA expression only at the targeted subtelomeres.

### 2.5. Demethylation of The Subtelomeric CpG Islands Induces An Increase in NRF1 Binding

NRF1 is a methyl-sensitive transcription factor possessing binding sites on each 29 bp repeat and this protein has been shown to regulate telomere transcription [22]. This prompted us to study the involvement of NRF1 in the DNA demethylation-dependent up-regulation of TERRA. To determine if methylation of the subtelomeric CpG islands influences NRF1 binding, we performed ChIP (chromatin immunoprecipitation) experiments. The large amount of chromatin required for the ChIP assays was difficult to obtain from transfected cells as the number of cells per transfection is limited. To cope with this, ChIP experiments were performed from HeLa cells untreated or treated with the DNA methyltransferase inhibitor 5-aza-dC for 72 h. For each of these two conditions, three biological replicates were used for ChIP assays. Semi-quantitative PCR was carried out using primers specific for the subtelomeric CpG islands (Xq, Yq, 15q, 16p, 9p), and PCR products were run on a BET-agarose gel. The ChIP primers were designed immediately downstream of the region containing tandem repeats since PCR amplification of this sequence was not possible. We observed a 2.3-fold higher signal when cells were treated with 5-aza-dC than when cells were untreated (Figure 3A), suggesting that demethylation induces an increase in NRF1 binding to these subtelomeric loci. A sequence from the *36B4* gene, which is not adjacent to any NRF1 binding sites, was amplified to determine the level of background. No difference in signal from the *36B4* gene was observed between treated and untreated samples when immunoprecipitation was performed with antibody against NRF1 (Figure 3A). 

To verify that these results were due to the DNA demethylation of the subtelomeres and not to indirect effects of the 5-aza-dC treatment, we examined NRF1 binding to a recognition sequence demethylated in untreated HeLa cells, in such a way that no difference of DNA methylation between untreated and 5-aza-dC-treated cells was expected at this site. We chose the unique NRF1 binding site located in the promoter of the *TFB2M* (transcription factor B2 of the mitochondria) gene [31], which is basically demethylated in HeLa cells according to whole-genome bisulfite sequencing [32]. No change in NRF1 binding to the *TFB2M* gene promoter was observed in 5-aza-dC-treated samples compared to the untreated samples (Figure 3A). These results demonstrate that when DNA methylation of an NRF1 binding site was not affected, the 5-aza-dC treatment did not induce an increase in NRF1 binding. Moreover, neither the mRNA level nor the protein level of NRF1 was affected by the 5-aza-dC treatment (Figure 3B). Together these results suggest that subtelomeric DNA demethylation induced an increase of NRF1 binding to subtelomeric sequences.

Of note, even for 5-aza-dC-treated samples, the normalized signal obtained with the anti-NRF1 antibody was much lower for subtelomeric CpG islands (Xq, Yq, 15q, 16p, 9p) than for the *TFB2M* promoter (0.91 and 7.94, respectively). This difference may be due to incomplete demethylation of the subtelomeres by 5-aza-dC (Appendix A), or it may be that NRF1 has a lower affinity for the subtelomeric binding sites than for the site located in the *TFB2M* promoter. In support of the latter hypothesis, compared to the NRF1 consensus binding site [33], two nucleotides are different in the binding sites located in 29 bp repeat regions, and only one nucleotide is different in the binding site located in the *TFB2M* promoter (Appendix A). Another possibility is that due to heterochromatic structure of the subtelomeric region [34], the subtelomeric NRF1 binding sites are hardly accessible.

### 2.6. NRF1 Is Involved in The DNA Demethylation-Dependent Up-Regulation of TERRA

To evaluate whether or not NRF1 is involved in the upregulation of TERRA induced by specific demethylation of subtelomeric DNA, we depleted HeLa cells transfected with sub gRNAs and the CRISPR-dCas9-TET1 system of NRF1 using small interfering RNAs (siRNAs). A siRNA designed to have no human transcript target (siLuci) was used as the negative control. The depletion of *NRF1* was confirmed by RT-qPCR and western blot (Figure 3C). When the sub gRNA and active TET1 were delivered in HeLa cells, silencing of NRF1 induced a significant decrease in TERRA compared to cells treated with the control siRNA (Figure 3D), suggesting that the up-regulation of TERRA induced by specific subtelomeric DNA demethylation is, at least partially, dependent on increased NRF1 binding to subtelomeres. Surprisingly, when HeLa cells were transfected with gRNAs targeting the subtelomere repeats and inactive TET1, *NRF1* silencing induced a significant increase of the TERRA level. This unexpected effect could be explained by the involvement of NRF1 in several extra-mitochondrial biological process and will be discussed below.

### 2.7. Compound C Induces An AMPK Inhibition-Independent up-Regulation of TERRA

The transactivation activity of NRF1 is promoted by the AMP-activated kinase (AMPK) pathway via the coactivator PGC-1α (peroxisome proliferator activated receptor gamma coactivator 1 alpha): Phosphorylation of PGC-1α by AMPK promotes its binding to NRF1, which increases the activity of this transcription factor [24,35]. To explore the potential regulation of TERRA expression from the subtelomeric CpG islands by the basal activity of the AMPK pathway, we treated HeLa cells with 6-[4-(2-piperidin-1-ylethoxy) phenyl]-3-pyridin-4-ylpyrazolo [1,5-a]pyrimidine (compound C), which is commonly used as a cell-permeable AMPK inhibitor [36]. Surprisingly, treatment with 5 µM compound C for 18 h induced a strong 21-fold increase in TERRA produced from subtelomeres Xq, Yq, 15q, and 9p in HeLa cells (Figure 4A). A significant increase in TERRA transcripts from these subtelomeres was also observed in compound C-treated HCT116 cells, although to a lower extent (Appendix A). To determine if the compound C treatment affected only the TERRA level from promoters with 61-29-37 repeats, we quantified TERRA from the subtelomeres Xp and Yp, which were previously demonstrated to be transcribed but do not contain the 61-29-37 repeats [4,10]. Treatment of HeLa cells with compound C induced a 4-fold increase in TERRA levels from subtelomeres Xp and Yp (Figure 4A), indicating that compound C induces an increase of the TERRA level independently of the sequence of TERRA promoter.

AMPK is activated when its subunit α is phosphorylated at threonine 172, and the carboxylase ACC (acetyl-CoA carboxylase) is one of the targets of this kinase [37]. To study the effect of the compound C treatment on the AMPK activity in HeLa cells, western blot analysis of the phosphorylation of AMPK and of ACC was performed. No decrease of the phosphorylation of AMPK or ACC was detected in compound C-treated cells compared to dimethyl sulfoxide (DMSO)-treated cells (Figure 4B), indicating that the compound C treatment did not inhibit AMPK activity in these experimental conditions. This suggests that the up-regulation of TERRA induced by compound C is not due to AMPK inhibition.

To determine if the up-regulation of TERRA driven by compound C was due to demethylation of the subtelomeric DNA, the methylation of the 29 bp repeats was analyzed. No change of the methylation status of the 29 bp repeats was observed in the HeLa cells after treatment with compound C (Figure 4C), indicating that compound C leads to a strong increase of the transcription from subtelomeres without altering their hypermethylated status.

## 3. Discussion

Previous studies have shown that the genetic or pharmacologic inactivation of DNA methyltransferases is associated with the demethylation of a CpG island promoter partially conserved on different subtelomeres and an increase of the TERRA level from these subtelomeres [10,11]. These results suggest that DNA methylation of the subtelomeric CpG islands modulates TERRA expression. Study of this phenomenon is complicated by the fact that methylation of the whole genome is affected by inactivation of the DNA methyltransferase activity and, consequently, indirect effects cannot be excluded [13,38]. CRISPR-Cas9 technology made it possible to develop an epigenetic engineering tool that can demethylate specific CpG sites [39]. To study how TERRA transcription is regulated by DNA methylation, we used a CRISPR-dCas9-TET1 system to specifically demethylate the subtelomeric CpG islands in HeLa cells.

These regions contain 29 bp repeats, which would play a central role in the activity of the TERRA promoter [10,27]. Using a method we developed that allowed us to analyze the methylation status of these critical repetitive elements, we observed a TET1 catalytic activity-dependent demethylation of the subtelomeric 29 bp repeats in HeLa cells transfected with the CRISPR-dCas9-TET1 system and RNA guides targeting the subtelomeric CpG islands. A control RNA guide did not cause demethylation of these regions indicating that the transfection with the CRIPSR-dCas9-TET1 system did not induce a global demethylation of the genome. Thus, this tool enables a specific demethylation of the targeted subtelomeric region, although off-target effects due to the binding of the RNA guides out of the targeted sites cannot be totally excluded.

We showed that the targeted demethylation of subtelomeric CpG islands was associated with an increase of the TERRA level. Then, to determine the underlying mechanism, we were interested in methyl-sensitive transcription factors previously described as being involved in TERRA regulation. A methyl-sensitive transcription factor CTCF (CCCTC-binding factor) was previously described as a positive regulator of TERRA [27,40]. Binding sites for this factor are located about 300 bp downstream of the subtelomeric 29 bp repeats. Although it has been shown that the CTCF binding to some promoters is DNA methylation dependent [41,42,43], Maurano et al. found that when DNA methyltransferases are depleted, CTCF binding is affected at a minority of sites [44]. Thus, most CTCF binding sites are methylation insensitive. In line with this, CTCF binding is affected by cytosine methylation at position 2 of the 15 bp long CTCF recognition sequence but not by cytosine methylation at position 12 [45]. The cytosine at the position 2 of the subtelomeric CTCF recognition site is followed by an adenine and not a guanine, making it likely that the binding of this transcription factor to subtelomeres is not affected by DNA methylation. The transcription factor NRF1 was also previously identified as a positive regulator of the TERRA transcription from the subtelomeres containing the CpG island [22]. Each 29 bp repeat contains two putative NRF1 binding sites, which could explain the importance of this repetitive tract in the activity of the TERRA promoter. It is well established that DNA methylation of the recognition site of NRF1 inhibits its binding [19,20,21,23,46]. We showed by ChIP that the treatment of HeLa cells with the DNA methyltransferase inhibitor 5-aza-dC resulted in an increase in NRF1 binding to the subtelomeric CpG islands. Moreover, in cells depleted of NRF1 by treatment with siRNA, TERRA up-regulation induced by specific demethylation of subtelomeric CpG islands was significantly attenuated. Altogether, these results suggest that the increase in NRF1 binding to subtelomeric CpG islands that results upon demethylation of these regions is involved in the TERRA up-regulation observed.

Understanding TERRA transcription regulation by subtelomeric DNA methylation is of special interest concerning the altered regulation of telomere length in pathologies, as cancer and ICF (immunodeficiency, centromeric instability and facial anomalies) syndrome. As mentioned above, two telomere length maintenance mechanisms have been described in cancer cells, telomerase activity, and ALT. Telomerase-positive and ALT cancer cells have differences in subtelomeric DNA methylation patterns and TERRA expression levels: telomerase-positive cells are characterized by hypermethylation of subtelomeric DNA and low levels of TERRA expression, whereas ALT cells generally present a lower methylation of subtelomeric DNA and a higher TERRA level compared to telomerase-positives cells [29]. In line with this, a negative correlation has been described across several cell lines between subtelomeric methylation and telomere recombination, a hallmark of ALT cells [47]. Interestingly, the difference of TERRA expression between these two types of cancer cells could be related to their respective telomere length maintenance mechanisms. In telomerase-positive cells, TERRA silencing would be a selective advantage because this transcript would inhibit telomerase activity, as suggested by in vivo and in vitro studies [9,48]. Nevertheless, other report has challenged this TERRA function [11]. In ALT cells, the recombinogenic nature of telomeres is essential for their maintenance and recombination depends on TERRA hybridization with telomeric DNA [8]. ICF syndrome type I is a rare autosomal-recessive disease caused by mutations in the *DNMT3B* gene. Cells from ICF1 patients are characterized by hypomethylation of their subtelomeres, elevated levels of TERRA, and short telomeres [12]. The accumulation of TERRA-telomere hybrids resulting from the high levels of TERRA appears to lead to the accelerated telomere shortening observed in this disease [49].

Although a negative correlation between TERRA expression and subtelomeric methylation has been described in both cancer cells and cells from ICF1 patients, regulation of TERRA by methylation of the subtelomeric CpG islands and the underlying mechanism had not been clearly established. Our results demonstrate subtelomeric DNA methylation dependent-TERRA regulation, reinforcing the hypothesis that there is a strong relationship between subtelomeric DNA methylation and telomere length dynamic via regulation of TERRA. Moreover, our data show that the transcription factor NRF1 is a key player in the DNA methylation-dependent regulation of TERRA. Our findings are in accordance with previous studies that showed that DNA methylation within NRF1 binding sites located in gene promoters is associated with inhibition of the binding of this protein and transcriptional repression [19,20,23,46]. Although NRF1 was first described as a transcription factor involved in mitochondrial biogenesis, recent work has shown that it participates in multiple biological functions such as splicing, cell cycle regulation, and DNA damage repair [50,51].

Because of its influence on TERRA expression, understanding dynamics of the subtelomeric DNA methylation during normal development and tumorigenesis is of special interest. Analyses of DNA methylation of several subtelomeric loci during these two processes have been previously performed. A subtelomeric locus present on 10% to 25% of chromosome ends is hypomethylated in gametes and undergoes a strong methylation during embryonic development [52]. In normal somatic cells of adults, around 80% of the cytosines of the subtelomeric CpG islands located on chromosome 2p, 4p, and 18p are methylated, and the methylation level does not seem to change during aging [29]. An increase of methylation in these subtelomeric loci was observed in telomerase-positive cell lines but not in ALT cell lines [29]. In gliomas and hepatocarcinomas, methylation is abnormal only on a subset of the subtelomeres analyzed [53,54]. Although informative, these studies focused on subtelomeric regions, whose role in TERRA expression has not been established. The importance of the 29 bp repeats in the promoter activity of subtelomeric CpG islands was demonstrated previously in reporter-based assays [10,27]. Our study suggests that, because of the presence of NRF1 binding sites on each 29 bp repeats, this repetitive tract plays a central role in the DNA methylation-dependent regulation of TERRA. The straightforward method we have developed for analysis of the methylation of CpGs present in these 29 bp repeats will enable study of the methylation pattern of this region during normal development and tumorigenesis.

Likewise, the mechanisms involved in the regulation of the methylation of the subtelomeric CpG islands still must be elucidated. The presence of a binding site for the CCCTC-binding factor (CTCF) at around 300 bp downstream of the 29 bp repeats [27] could be involved in this regulation. Indeed, several reports have suggested that this multifunctional transcription factor may induce a local demethylation of the cytosines in the vicinity of its binding site [55,56,57]. This property would be due to an interaction of CTCF with the DNA demethylases TET1 and TET2 [58]. Moreover, a previous genome-wide study showed that, in normal cells, the binding of transcription factors, including NRF1, to their consensus motif is promoted by the presence of other consensus motif in their vicinity, and this binding might protect neighboring regions from hypermethylation during tumorigenesis [59]. Thus, the subtelomeric CpG islands could be differentially targeted for hypermethylation during tumorigenesis depending on the combination of consensus motifs that they contain.

Diman et al. [22] have previously shown that pharmacological activation of the AMPK pathway induced TERRA upregulation. By exploring the regulation of telomere transcription by the AMPK pathway, we discovered an unexpected effect of compound C, which was identified as a reversible AMPK inhibitor that competes for the ATP (adenosine triphosphate) binding site [36]. We observed that treatment of HeLa cells with 5 µM compound C caused a strong increase of the TERRA levels from two different promoter types: the promoter type located on subtelomeres Xq, Yq, 15q, and 9p, which are associated with 61-29-37 repeat tracts, and the promoter type located on subtelomeres Xp and Yp, which lack this tract. Thus, compound C affected TERRA levels from subtelomeres with distinct promoter sequences, suggesting that this effect does not involve a sequence-specific regulator like a transcription factor but rather would be the result of a change affecting chromatin features of chromosome ends, shelterin assembly or TERRA processing. The concentration of compound C used in our study did not induce inhibition of AMPK activity, indicating that the increase of the TERRA level was AMPK inhibition-independent. AMPK inhibition-independent effects of compound C have been reported including induction of autophagy [60], anti-proliferative effects in glioma [61], and inhibition of the nonsense-mediated RNA decay [62]. Of note, the depletion of factors involved in the nonsense-mediated RNA decay pathway was previously associated with an accumulation of TERRA at the telomeres but no change of the TERRA level was detected [4], suggesting that the increase of the TERRA level induced by compound C is not dependent of the inhibition of this pathway. These multiple AMPK inhibition-independent effects of compound C are in line with the lack of specificity of this inhibitor [63]. Indeed, compound C inhibits kinases other than AMPK with an equal or better efficiency [64]. The kinases strongly inhibited by compound C include ERK8, MNK1, PHK, MELK, DYRK1A, DYRK2, DYRK3, HIPK2, Src, and Lck. One of these kinases could be involved in the expression or the degradation of TERRA. The elucidation of the mechanism involved in the increase of TERRA level induced by compound C could lead to identify new regulator of TERRA. 

In conclusion, in this study we showed that targeted DNA demethylation of the subtelomeric CpG island induces an increase of TERRA expression and this upregulation involves NRF1. Thus, transcriptional repression of TERRA by DNA methylation is notably due to the inhibition of NRF1 binding on the subtelomeric promoter of TERRA.

## 4. Materials and Methods

### 4.1. Construction of Plasmids

The gRNA vectors were generated by inserting the target sequences into pBlueScript-U6sgRNA (Addgene plasmid 43860) digested with BsmB1. Ligation of the gRNA insert fragment was performed using T4 DNA ligase (New England BioLabs, Ipswich, MA, USA). The site-directed mutagenesis of the scFv-GFP-TET1CD vector (Addgene plasmid 82561; H1671Y and D1673A mutations) was performed as follows: The plasmid was digested with NotI and NheI, and the two fragments were run on an agarose gel and purified using the EZNA Gel Extraction Kit (Omega, Norcross, GA, USA). The fragment containing the coding sequence of *TET1* served as template for two PCR with either assemb-TET1-F/TET1-mut-R oligonucleotides or assemb-TET1-R/TET1-mut-F oligonucleotides and using Phusion DNA polymerase (New England BioLabs). TET1-mut-R and TET1-mut-F are overlapping oligonucleotides with the desired mutations. The two PCR products were purified using the EZNA Cycle Pure Kit (Omega), and PCR was performed using a mix of these two PCR products and assemb-TET1-F/assemb-TET1-R oligonucleotides. Samples were run on an agarose gel, and the PCR product containing the mutated coding sequence of *TET1* was purified using the EZNA Gel Extraction Kit (Omega). This DNA molecule was inserted in the Addgene plasmid 82561 with the coding sequence of *TET1* deleted after digestion with NotI and NheI by Gibson assembly (New England BioLabs) following the manufacturer’s protocol. Cloning was performed in bacteria DH5 (New England BioLabs), and plasmid was harvested using EZNA Endo-free Plasmid DNA Mini Kit (Omega). Sanger sequencing (Eurofins, Nantes, France) was performed to confirm the mutation. 

### 4.2. Cell Culture, Transfections, and Treatments

The cell lines were purchased from ATCC. HeLa (human cervix carcinoma) and T98G (human glioblastoma) cell lines were cultured in DMEM, high glucose, GlutaMAX™ Supplement (ThermoFisher, Waltham, MA, USA) supplemented with 10% fetal calf serum and penicillin-streptomycin. HCT116 (human colorectal carcinoma), U2OS (human osteosarcoma), and Saos-2 (human osteosarcoma) cell lines were cultured in McCoy’s 5A (Modified) Medium, GlutaMAX™ Supplement (ThermoFisher) supplemented with 10% fetal calf serum and penicillin-streptomycin. Plasmid transfections of HeLa cells were performed with Amaxa Nucleofector II (Lonza, Basel, Switzerland) by using kit R (Lonza), program I-013. For each experiment, 1 × 10^6^ HeLa cells were transfected with 500 ng of the dCas9-peptide array fusion vector (Addgene plasmid 82560), 900 ng of the scFv-GFP-TET1CD vector (Addgene plasmid 82561), and 650 ng of each of the gRNA vectors (sub1, sub2, sub3). Transfections with siRNAs were performed using Lipofectamine 2000 (ThermoFisher) and using either 10 pmol of each siNRF1 (siNRF1-1, siNRF1-2, siNRF1-3) or 30 pmol of siLuci. Cell treatment with compound C (Sigma-Aldrich, Saint-Louis, MO, USA) was performed as follows: 5 × 10^5^ cells were seeded in 6-well culture plates (Eppendorf, Hamburg, Germany); 5 h later, culture medium was changed with 2 mL of new culture medium containing 5 µM compound C or DMSO, and cells were harvested 18 h later. Cell treatment with 5-aza-dC (Sigma) was performed as follows: HeLa cells diluted at 25% confluence at day 0 in a 15-cm plate were treated or not with 10 µM 5-aza-dC from day 1 to day 4 and harvested at day 4.

### 4.3. DNA Methylation Analysis

The digestion of genomic DNA was performed in 200 µL PCR tubes in which 500 ng or 1 µg of genomic DNA, 1 µL of HpaII (methyl-sensitive) or MspI (methyl-insensitive), 1.7 µL of CutSmart Buffer (New England BioLabs), and H_2_O up to 17 µL were mixed. Samples were incubated at 37 °C for 3 h and then at 80 °C for 20 min to inactivate endonucleases. Subsequently, 660 pmol of dNTPs, 0.1 pmol of OT oligonucleotide, 0.3 µL of CutSmart Buffer, and H_2_O up to 19 µL were added. Samples were placed in a thermocycler (Bio-Rad, Hercules, CA, USA), and denaturation of the genomic DNA was performed by heating (95 °C for 3 min) immediately followed by slow cooling (57 °C for 5 min, 55 °C for 5 min, 50 °C for 5 min, 45 °C for 5 min, 40 °C for 5 min) to allow hybridization of OT. The elongation of the DNA molecules hybridized to OT was performed using the modified Klenow fragment lacking 3′→5′ exonuclease activity (New England BioLabs) in order to avoid removal of the dideoxycytidine located at the 3′ end of OT. OT was blocked at its 3′ end by dideoxycytidines to prevent its elongation, which would produce false positive results. To perform elongation of the DNA molecules hybridized to OT, 1 µL of Klenow lacking exonuclease activity was added, and samples were incubated for 30 min at 37 °C and then at 75 °C for 20 min to inactivate the polymerase. The PCR with PR and PF oligonucleotides was performed as follows: 1 µL of the sample from elongation with OT was mixed with 5 pmol of each oligonucleotide, 4 nmol of dNTPs, 2 µL of PCR Reaction Buffer (20 mM MgCl_2_ 10× concentrated), and 0.16 µL of FastStart Taq DNA Polymerase (Merck, Darmstadt, Germany). Reactions were carried out using the following conditions: an initial step of 4 min at 95 °C, followed by a variable number of cycles of 30 s at 95 °C, 30 s at 58 °C, 15 s at 72 °C, and a final extension step of 7 min at 72 °C. PCR products were run on 1.5 % agarose gel, and the DNA bands were purified using the EZNA Gel Extraction Kit (Omega). Sanger sequencing (Eurofins) confirmed that the expected PCR products were obtained. The sequences of primers are listed in Table 1.

### 4.4. RNA and DNA Extraction

Phenol chloroform extraction was used to preserve RNA integrity. Cells were trypsined and centrifuged at 200 × g for 10 min at 4 °C. Cell pellets were suspended in 1 mL of TRIzol™ Reagent (ThermoFisher), and 200 µL of chloroform were added. Samples were vortexed and centrifuged at 12000 × g for 15 min at 4 °C. The aqueous phase was collected, and RNA was precipitated with 1 volume of isopropanol. The pellets were washed with 70% ethanol, and RNA was dissolved in 50 µL H_2_O. To remove DNA contamination, an extensive DNase I treatment was performed. For this treatment, RNA was treated with 2 µL RNase-free DNase I (New England Biolabs) in a 50-µL volume including 5 µL 10X DNase I Reaction buffer, 1 µL RNase inhibitor, Murine (New England Biolabs), and H_2_O for 30 min at 37 °C. RNA was precipitated with Lithium Chloride Precipitation Solution (Invitrogen) according to the manufacturer’s protocol, and resuspended in 30 µL H_2_O. Genomic DNA extraction was performed using EZNA Tissue DNA kit (Omega) according to the manufacturer’s protocol. RNA and DNA were quantified by UV spectrometry using NanoDrop 1000 spectrophotometer (ThermoFisher).

### 4.5. Quantitative RT-PCR Analysis

Reverse transcription of TERRA was performed in 200-µL PCR tubes in which 5 µg RNA, 2 pmol of RT-TERRA primer, 10 pmol of GAPDH-F primer, 10 nmol of dNTPs, and H_2_O up to a final 13 µL volume were mixed. The samples were incubated in a PCR thermocycler (Bio-Rad) at 65 °C for 5 min, transferred to ice, and 4 µL of 5X First Strand Buffer, 1 µL 0.1 M DTT, 1 µL RNase inhibitor, Murine (New England Biolabs), and 1 µL 200 U/µL SuperScript III RT (ThermoFisher) or H_2_O for no-RT control were added. The samples were incubated in a PCR thermocycler at 55 °C for 60 min followed by enzyme inactivation at 70 °C for 15 min. The reverse transcription of NRF1 was performed in the same conditions using the NRF1-R primer.

Quantitative PCR was performed in an Mx3000P qPCR System (Agilent, Santa Clara, CA, USA) using Brilliant II SYBR^®^ Green qPCR Master Mix (Agilent) in a 96-well reaction plate (Agilent). For each qPCR sample two technical replicates were carried out. For qPCRs with TR/TF, GAPDH-R/GAPDH-F, and NRF1-R/NRF1-F, each reaction mix contained 1 µL of reverse transcription product, 5 pmol of each primer, and 1X Brilliant II SYBR^®^ Green qPCR Master Mix (Agilent) in a total volume of 20 µL. The plates were sealed with optically clear Strip Caps (Agilent). The qPCRs with TR/TF, GAPDH-R/GAPDH-F, and NRF1-R/NRF1-F were performed according to the following program: 1 cycle of denaturation at 95 °C for 15 min, 40 cycles of denaturation at 95 °C (10 s), annealing/extension at 60 °C (30 s), and dissociation for melting curve analysis. For each primer pair, a single peak was observed in dissociation curves, indicating amplification of a single amplicon (data not shown). Relative changes in TERRA and *NRF1* levels between samples were determined using the 2-ΔΔCt method with *GAPDH* as reference gene as described previously [65].

### 4.6. Protein Extraction and Western Blot Analysis

Total protein extracts were prepared as follows: Cells were trypsined and washed with cold DPBS 1X (ThermoFisher) and then were centrifuged at 200 × g for 10 min at 4 °C. Cell pellets were suspended in 50 µL of ice-cold lysis buffer [100 mM Tris-HCl, pH 7.4, 1% Triton, 0.25% sodium deoxycholate, 0.1% SDS, 300 mM NaCl, 1 mM EDTA, 1X PhosSTOP (Sigma), 1X Protease Inhibitor Cocktail (Sigma)]. Protein concentrations were determined using the Pierce BCA Protein Assay Kit (ThermoFisher). Migration of an equal mass of total protein and of 5 µL PageRuler Plus Prestained Protein Ladder, 10 to 250 kDa (ThermoFisher) was carried out on an 8% acrylamide denaturing gel. Proteins were transferred onto a nitrocellulose membrane using Trans-Blot Turbo Transfer System (Bio-Rad, city, state abbreviation if USA or Canada, country). The primary antibodies used against NRF1, p-AMPK, p-ACC and tubulin were mouse ab55744 (Abcam, Cambridge, United Kingdom), rabbit #07-681 (Merck), rabbit #11818 (Cell Signaling, Danvers, MA, USA), and mouse T5168 (Sigma), respectively. Incubation of the membrane with the primary antibodies was performed according to the manufacturer’s protocol. The secondary antibodies used were goat anti-rabbit #65-6120 and goat anti-mouse #62-6520 (Invitrogen). Detection of proteins was performed by chemiluminescence using Clarity Western ECL Blotting Substrate (Bio-Rad).

### 4.7. ChIP Assay and Semi-Quantitative PCR Analysis

ChIP was performed using ChIP-IT Express (Active Motif, Carlsbad, CA, USA) according to the manufacturer’s protocol. Briefly, HeLa cells untreated or treated with 5-aza-dC (10 µM) for 3 days were harvested after crosslinking with formaldehyde. Cells were lysed and the nuclei were sonicated by performing 10 cycles (30 s on—30 s off) in the Bioruptor device (Diagenode, Liège, Belgium) with intensity set to the mode “high”. The immunoprecipitations were carried out using 25 µg of sheared chromatin and 3 µg of rabbit anti-NRF1 antibody #ab34682 (Abcam) or 3 µg of non-specific rabbit IgG #C15410206 (Diagenode). Semi-quantitative PCR was performed by mixing 1/50 of the immuno-precipitated samples (2 µL) or 1 ng of chromatin from the Input samples, 5 pmol of each oligonucleotide, 4 nmol of dNTPs, 2 µL of PCR Reaction Buffer (20 mM MgCl_2_, 10× concentrated), and 0.16 µL of FastStart Taq DNA Polymerase (Merck). Linearity of the PCR amplifications was established experimentally. For CHIP-F/CHIP-R the following conditions were used: an initial step of 4 min at 95 °C, followed by 35 cycles of 30 s at 95°C, 30 s at 62 °C, 30 s at 72 °C, and a final extension step of 5 min at 72 °C. For TFB2M-F/TFB2M-R the following conditions were used: an initial step of 4 min at 95 °C, followed by 31 cycles of 30 s at 95 °C, 30 s at 59 °C, 30 s at 72 °C, and a final extension step of 5 min at 72 °C. For 36B4-F/36B4-R the following conditions were used: an initial step of 10 min at 95 °C, followed by 32 cycles of 30 s at 95 °C, 1 min s at 60 °C. PCR products were run on 1.5% agarose gel and stained with ethidium bromide (BET). Signal quantification was done using ImageJ software (National Institutes of Health, Bethesda, MD, USA).

## Figures and Tables

**Figure 1 ijms-20-02791-f001:**
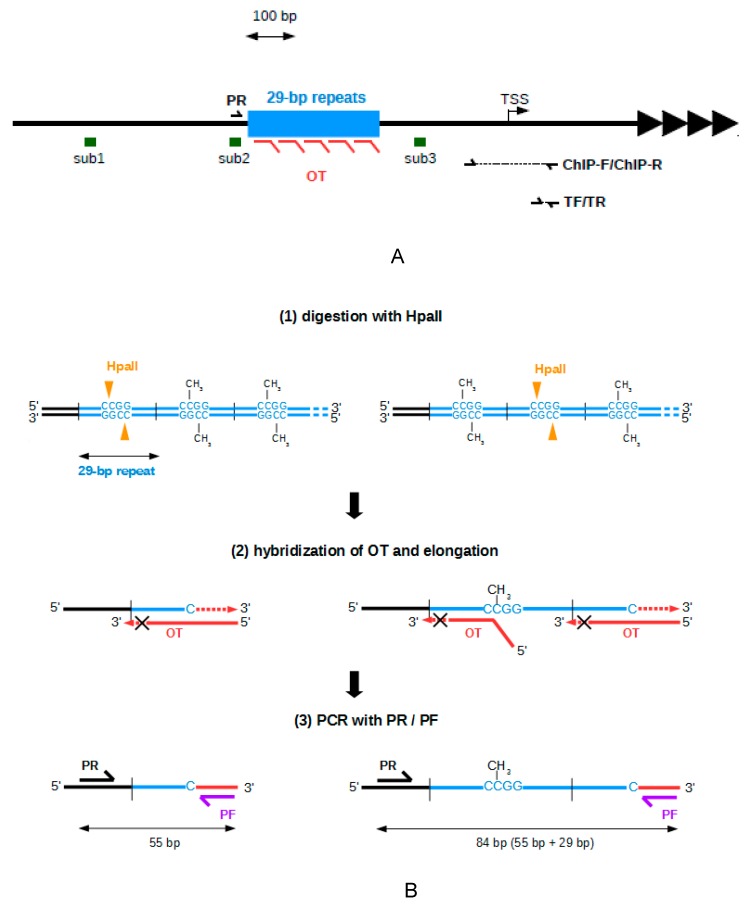
The 29 bp repeats of the subtelomeric CpG islands are hypermethylated in HeLa cells. (**A**) Schematic representation of the subtelomeric region containing CpG island. Arrowheads indicate telomeric repeats, and blue box represents the subtelomeric 29 bp repeats. The subtelomeric telomere repeat containing RNA (TERRA) transcription start site (TSS) is indicated by a black arrow. The locations of the targets for RNA guides (sub1, sub2, and sub3) are indicated by green bars. The binding sites of the Reverse Primer (PR), Oligonucleotide Template (OT) (red), and the primer pairs used for the chromatin immunoprecipitation (ChIP) (ChIP-Foward/ChIP-Reverse (ChiP-F, ChiP-R)) and primers for the qPCR of TERRA, Foward (TF) and Reverse (TR) are indicated. Sequences amplified by these two latter primer pairs are indicated by dashed lines. (**B**) Schematic representation of the application of the method of analysis of the cytosine methylation at 29 bp repeats. In this example, two DNA molecules with two different patterns of cytosine methylation in the 29 bp repeats are represented (blue lines). Other 29 bp repeats are represented by dotted blue lines. (1) Genomic DNA is digested by the methylation-sensitive endonuclease HpaII (the HpaII recognition sites located within each 29 bp repeat are indicated by orange arrowheads). (2) After hybridization of the 3′ region of the oligonucleotide OT (red line) to the sequence adjacent to the HpaII recognition sites of the 29 bp repeats, DNA polymerase-catalyzed elongation (represented by dotted red lines) is performed in which the 5′ region of OT serves as template. OT is blocked at its 3′ end by dideoxycytidines to prevent its own elongation represented by crossed dotted red lines. (3) PCR is carried out with the Reverse Primer PR (black, complementary to a region upstream of the 29 bp repeats) and Forward Primer (PF) (purple, complementary to sequence added at the 3′ end of the cleaved DNA molecules). In this example, PCR products of two different sizes are obtained depending of the methylation pattern of the region. (**C**) Validation of the method was performed by analysis of 1 µg of genomic DNA extracted from HeLa cells untreated or treated with 5-aza-dC was undigested or digested by HpaII (methyl sensitive) or MspI (methyl insensitive). A sample without DNA (blank) was used as negative control. Elongation products (1/20 of the reaction) were amplified by PCR (35 PCR cycles). PCR products were run on a 1.5% BET-agarose gel. Ladder is the 50 bp DNA ladder (New England BioLabs).

**Figure 2 ijms-20-02791-f002:**
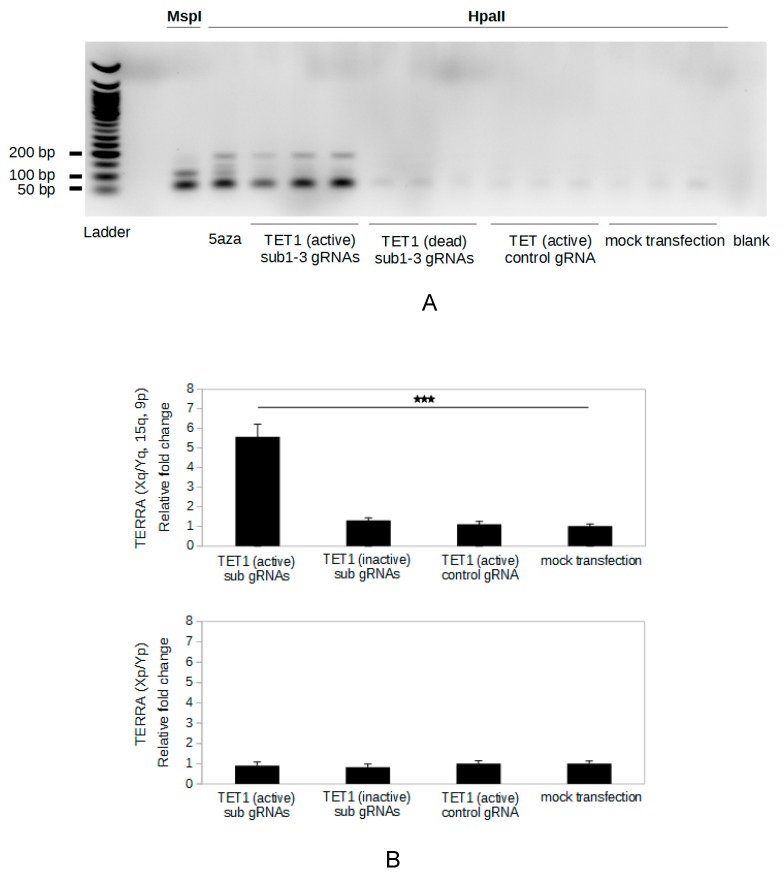
Demethylation of the subtelomeric 29 bp repeats by the CRISPR-dCas9-TET1 system is associated with an up-regulation of TERRA. HeLa cells were transfected with the CRISPR-dCas9-TET1 system with TET1 active or catalytically dead and subtelomeric gRNAs (sub1-3) or control gRNA. Genomic DNA and RNA were extracted 96 h post-transfection. Mock transfected HeLa cells were used as control. (**A**) The methylation of the 29 bp repeats was analyzed using the method described in Figure 1B: 1 µg of genomic DNA was digested with HpaII, 1/20 of the elongation reaction was amplified by PCR (35 PCR cycles), and PCR products were run on a 1.5% bromure ethidium (BET)-agarose gel. Three independent transfection experiments were done for each condition. Genomic DNA from HeLa cells treated with 5-aza-dC and digested by HpaII (5aza) or untreated and digested with MspI were used as positive controls. A sample without DNA (blank) was used as negative control. Ladder is the 50 bp DNA ladder (New England BioLabs). (**B**) TERRA from subtelomeres Xq, Yq, 15q, 9p and TERRA from subtelomeres Xp and Yp were quantified by RT-qPCR (reverse transcription-quantitative PCR) using TF/TR and Xp-R/Xp-F primer pairs, respectively. Levels were normalized to *GAPDH* (glyceraldehyde-3-phosphate dehydrogenase) mRNA, and all values were compared to mock transfected sample. The bars are average values from three biological and two technical replicates for each sample. Error bars are standard deviations. *p* values were calculated by paired two-tailed Student’s t-test (*n* = 3). *** *p* < 0.001.

**Figure 3 ijms-20-02791-f003:**
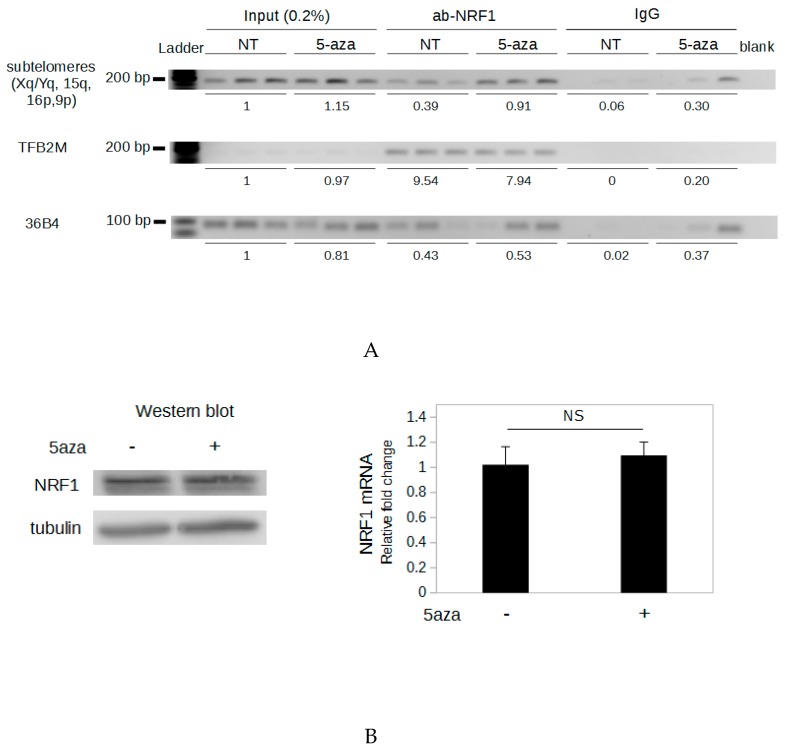
NRF1 (nuclear respiratory factor 1) binding to subtelomeres is involved in DNA methylation-dependent regulation of TERRA expression. (**A**) Chromatin immunoprecipitation (ChIP) assays were performed using chromatin extracted from HeLa cells untreated (NT) or treated with 10 µM 5-aza-dC (5aza) for 72 h and an antibody that recognizes NRF1 (ab-NRF1) or a non-specific antibody (IgG). Primers specific for subtelomeres Xq, Yq, 15q, 16p, and 9p (ChIP-F/ChIP-R), the promoter region of the *TFB2M* (transcription factor B2 of the mitochondria) gene (TFB2M-F/TFB2M-R), or a region of the *36B4* gene devoid of NRF1 binding sites (36B4-F/36B4-R) were used to detect co-precipitated chromatin fragments. For each condition (untreated and 5-aza-dC-treated), three biological replicates were used for the ChIP assay. Each lane corresponds to a biological replicate. Values above the gel correspond to the average signal intensity normalized to the Input of untreated cells. Ladder is the 50 bp DNA ladder (New England BioLabs). (**B**) HeLa cells were untreated (−) or treated (+) with 10 µM 5-aza-dC (5aza) for 72 h. **Left***,* Western blot analysis of NRF1 protein. Tubulin was used as a loading control; 30 µg of total protein was loaded in each well. **Right**, RT-qPCR analysis of *NRF1* mRNA. Levels were normalized to *GAPDH* mRNA, and all values were compared to untreated cells. (**C**) HeLa cells were lipofected with small interfering RNAs (siRNAs) targeting *NRF1* (siNRF1) or a control siRNA targeting *luciferase* (siLuci). Lipofections were performed at days 1 and 2, and proteins and RNA were extracted at day 3. **Left**, Western blot analysis of NRF1 protein. Tubulin was used as a loading control; 30 µg of total protein was loaded in each well. **Right**, RT-qPCR analysis of the *NRF1* mRNA. Levels were normalized to *GAPDH* mRNA, and all values were compared to cells lipofected with siLuci. (**D**) RT-qPCR analysis of TERRA produced from Xq, Yq, 15q, and 9p in HeLa cells transfected with the CRISPR-dCas9-TET1 system (TET1 active or dead) and subtelomeric gRNAs then lipofected with siNRF1 or siLuci. Transfection was performed at day 0, lipofections were performed at days 1 and 2, and RNA was extracted at day 3. TERRA was quantified by RT-qPCR using TR and TF; levels were normalized to *GAPDH* mRNA, and all values were compared to the samples transfected with catalytically dead TET1 and lipofected with siLuci. The bars represent the average values from three biological and two technical replicates for each sample. Error bars represent the standard deviations. *p* values were calculated by paired two-tailed Student’s t-test (*n* = 3). * *p* < 0.05. ** *p* < 0.01. NS: not significant (*p* > 0.05).

**Figure 4 ijms-20-02791-f004:**
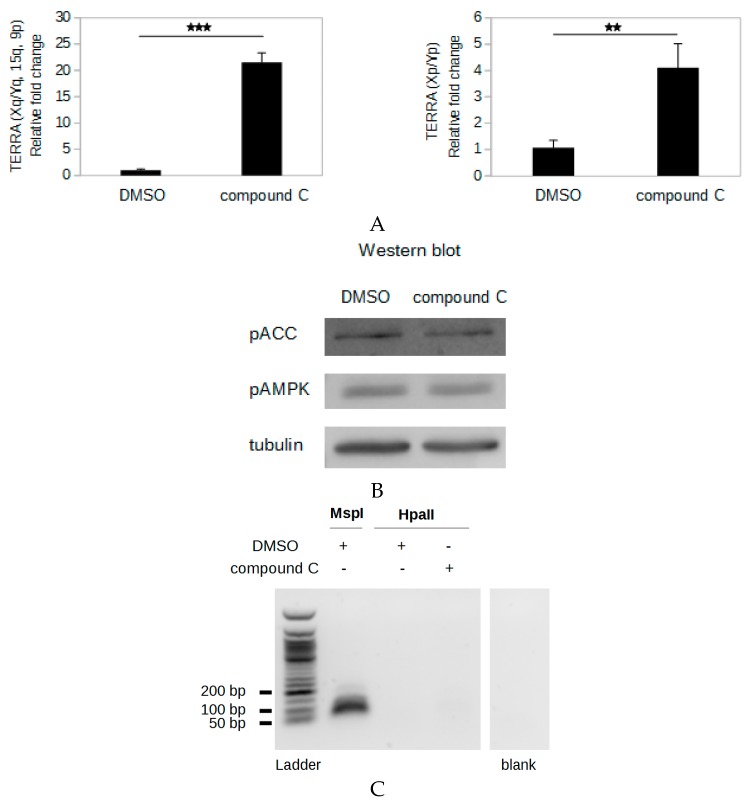
Compound C induces an increase in TERRA levels in HeLa cells. HeLa cells were treated with dimethyl sulfoxide (DMSO) or compound C (5 µM). DNA, RNA, and proteins were extracted after 18 h of treatment. (**A**) TERRA from subtelomeres Xq, Yq, 15q, and 9p and TERRA from subtelomeres Xp and Yp was quantified by RT-qPCR using TR/TF and Xp-R/Xp-F primer pairs, respectively; levels were normalized to *GAPDH* mRNA, and all values were compared to DMSO-treated sample. The bars represent the average values from three biological and two technical replicates for each sample. Error bars represent the standard deviations. *p* values were calculated by paired two-tailed Student’s t-test (*n* = 3). ** *p* < 0.01. *** *p* < 0.001. (**B**) Western blot analysis of the phosphorylation of AMP-activated kinase (AMPK) and acetyl-CoA carboxylase (ACC). Tubulin was used as a loading control; 15 µg of total protein was loaded in each well. This result is representative of five independent experiments. (**C**) Analysis of the methylation of the 29 bp repeats using the method described in Figure 1B. Genomic DNA (500 ng) was digested with HpaII (methyl sensitive) or MspI (methyl insensitive). A sample without DNA (blank) was used as negative control. Elongation products (1/20 of sample) were amplified by PCR (35 PCR cycles). PCR products were run on a 1.5% BET-agarose gel. Ladder is the 50 bp DNA ladder (New England BioLabs).

**Table 1 ijms-20-02791-t001:** Oligonucleotide sequences.

Oligonucleotide	Sequence (5′-3′)
assemb-TET1-F	CGGACCGGTGGCGGTGGCGGAGGGGCTAGCAGATCCGAACTGCCCACCTGCAGC
assemb-TET1-R	ATGGCTGATTATGATCTAGAGTCGCGGCCGCTCAGACCCAATGGTTATAGGG
TET1-mut-R	TTGTGAATGGCCCTGTAGGGATGAGCACAGAAGTCC
TET1-mut-F	CTGTGCTCATCCCTACAGGGCCATTCACAACATGAATAATGGAAG
TF	GCAGCCATGAATAATCAAGGT
TR	TTCCGCACTGAACCGCTCTAA
ChIP-F	GCTCTAACTGGTCTCTGACCT
ChIP-R	GTTCTGCTCAGCACAGACCTG
TFB2M-R	ACGGTCCACTCACAATCCTC
TFB2M-F	CCCACGTGGAACATTTTCTG
36B4-F	CAGCAAGTGGGAAGGTGTAATCC
36B4-R	CCCATTCTATCATCAACGGGTACAA
Xp-R	TCTTCTTTCTGGTGGGGTTG
Xp-F	GGGGTCCCTTTCCATACTGT
GAPDH-R	GAAGGTGAAGGTCGGAGTCAAC
GAPDH-F	CAGAGTTAAAAGCAGCCCTGGT
OT	GTTAGATCCCAGGCGTAGAACAGGCGCAGGCGCAGAGATddC
PR	GGACGCGCTAGCATGTGT
PF	GTTAGATCCCAGGCGTAGAACAG
sub1	GCAGGGCTCTCTTGCTTAGAG
sub2	CCTGCGCCACGCCTCCACCCC
sub3	GTCCTCTGCACAGATTTCGG

ddC: dideoxycytidine. GAPDH:

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
