# Peer review of "Repression of TERRA Expression by Subtelomeric DNA Methylation Is Dependent on NRF1 Binding"

_ijms, 2019, doi:10.3390/ijms20112791_

Reviewer 1 Report

In this manuscript  Le Berre and coworkers study the influence of methylation of subtelomeric 29-bp repeats on the level of TERRA (telomeric repeat containing RNA).  For this they use the CRISPR-dCas9-Tet1 system to locally demethylate DNA and a novel MSRE-PCR-based method to assess the level of 29-bp repeat methylation. Using chromatin immunoprecipitation they also assess NRF1 binding to methylated and demethylated (by treatment with 5-azacytidine) sequence in TERRA promoter. Furthermore they investigate the effect of NRF1 expression silencing by siRNA on TERRA expression. Finally, the Authors investigate the effect of compound C, an inhibitor of AMPK, on TERRA expression.

This is an interesting research in a relatively novel area concerning transcription of telomeric DNA and the effect of CpG methylation in  subtelomeric promoter sequences on this transcription. I have several concerns that, in my opinion,  should be  addressed before publication.

Major:

1. Although the principle of the MSRE-PCR-based method developed by the Authors to assess the extent of methylation of the 29-bp repeats seems sound, the results should be compared to those obtained by the “ classical” MSRE-PCR method. That is, a supplementary figure should show the pattern/intensity of PCR products obtained by both methods, possibly in both  untreated and 5-azacytidine treated cells. Such comparison would also allow the Authors to demonstrate the advantages of using the novel method.

2. Although the Authors state that NRF1 possesses binding sites on each 29-bp repeat and they specifically assessed demethylation of these repeats, the binding of NRF1 was investigated in another area as can be judged by the location of ChIP primers. This should be explained/corrected.

3. The part concerning the effect of compound C is the least consistent and should be either omitted or further elaborated. Since this compound is not a specific AMPK inhibitor another, more specific inhibitor or, alternatively, activator, of AMPK should be tested to exclude involvement of this kinase in NRF1 activation . Also, based on blots in Fig. 4B, the Authors state that “compound C treatment did not inhibit AMPK activity in these experimental conditions”.  To state this the Authors should perform a quantitative analysis of data  from several experiments; a single blot result is not enough to draw definite conclusion especially that 5 micromolar concentration of this drug proved effective in inhibiting AMPK in other studies.

Minor:

4. The length of the lower bands of the Ladder should be given in Figs 1C, 2A etc., so that the  length of the PCR products could be evaluated independently of the description in the text.

5. Line 272 and Fig.3BC – should be Nrf1 mRNA

Several typographical and/or language errors should be corrected, e.g. Lines 52 - methyltransferases, 403 -interested in, 461-undergoes, 475-has to be elucidated.

Author Response
1. The MSRE-PCR method developed in this paper was modified so that a PCR amplification product would occur if the recognition site of the endonuclease HpaII, present in each of the 29-bp repeats, is demethylated. With the "classical" MSRE-PCR method no PCR product would be detected when the 29-bp repeats is demethylated, and for each methylation pattern of the 29-bp repeats several PCR products would be produced. Therefore it is not possible to compare the two methods. This point has been clarified in the results section 2.1 lines 152-154, 160-161.

2. The ChIP primers were designed immediately downstream of the region containing tandem repeats since PCR amplification of this region was not possible. This has been clarified in the results section 2.5 lines 276-277.

3. There are no another AMPK inhibitors described in the literature and activators of AMPK have already been tested by Diman et al (22). This has been added in the discussion.  lines 525-526

We agree with the reviewer recommendation and we performed 4 independent experiments to carry out a quantitative analysis of western-blots. Results are presented below and this has been mentioned in the figure 4 B caption.
4. The lengths of the bands have been added in Figure 1C, Figure 2A, Figure 4C.

5. These errors have been corrected

Western-Blot: quantitavive analysis

Tubulin was used for normalization. Bars represent the average values for four biological replicates. Error bars represent the standard deviations. P values were calculated by Student's t-test (n=4).

Reviewer 2 Report

The manuscript "Repression of TERRA expression by subtelomeric DNA methylation is dependent on NRF1 binding" by Le Berre and colleagues deals with the interesting question how long noncoding telomere repeat containing RNAs (TERRA) are regulated, in particular with respect to DNA methylation. TERRA transcripts are transcribed from transcription start sites in subtelomeres, the regions adjacent to the chromosome ends (telomeres). In human cells, TERRA promoters of many subtelomeres are rich in CpG and have three conserved repetitive DNA elements of specific lengths (therefore called 61-29-37 according to the lengths of these repeats). These CpG islands at these subtelomeres are highly methylated in telomerase-positive cell lines. However, the specific epigenetic regulation at these sites is not clear yet. The present study shed light on this important question.
There are, in my opinion, just a few points which are indicated below that need some clarification or re-ordering. In particular for readers who are not from the field, it might be a bit difficult to follow the main thread in the manuscript. I feel some sections could be streamlined and shortened to make the take-home messages clearer. In contrast, at some positions, some more details could be helpful. In addition, I recommend to add a short paragraph with a main conclusion. Below I give some detailed comments which I hope help to improve the manuscript.
I will go through the sections one by one.

# Abstract
1) It may sound like a totally trivial point but I suggest to mention the 'study organism' at least once either in the title or the first half of the abstract. While it becomes obvious that we are dealing with human cells, it could be stated clearly.
2) Lines 9-10: "long noncoding RNA called TERRA (for telomeric repeat-containing RNAs" - I recommend to re-write this sentence to first explain the abbreviation and then use it. Perhaps something like: Chromosome ends are transcribed into long noncoding telomere repeat containing RNAs (TERRA).
3) Line 20: What is compound C? It could be useful to explain it in the abstract. I see that the complete discription is an awful term of biochemistry, so another way would be to skip the word 'compond C' in the abstract and only speak of 'treatment with an AMPK inhibitor' or just reverse the sentence to 'we treated the cells with an AMPK inhibitor (compound C) if to everyone in the authors' field it is totally clear what it is (just not to me..). Currently, the authors still mix 'compound C' and 'Compound C' throughout the manuscript, I suggest to make the writing consistent.

# Introduction
4) Line 30: What is the shelterin complex? I would appreciate a short summary/description or a reference, for example de Lange (2005), Genes & Dev. To me, it is not 'textbook knowledge'.
5) Line 43: What are 'telomerase-positive cells'? I assume, the authors mean cancer cells here, but also normal stem cells could be meant (or both). I suggest to make this point clearer.
6) Line 45: Which specific human cell lines are the authors speaking about? Should be stated.
7) Lines 84-88: Could the authors comment on how specific the treatment with 5-aza-dC was expected to be? Or, if it was not necessary to be very specific, why side effects might not be important. I could imagine that a treatment of the whole cell line sample could have general effects on the gene expression/DNA methylation of different genes, not just the subtelomeres.

# Results
8) Lines 101-122 (Figure 1, caption): I really appreciate the elaborate form of the figure caption. I am just a bit confused about the 'abbreviations' (TF, TR, PR, OT..). It becomes somewhat clear later that we are dealing with oligonucleotides here, but I still have trouble to understand the mechanism at play. Perhaps the authors can re-order there information to make clearer what they test here. In addition, there is no indication of the size for the different bands in Figure 1C or which ladder was used (also in the following gel pictures, e.g. 2B). This information should be added to make the gel pictures more meaningful.
9) Lines 129-136: As I am not an expert in laboratory methods, I got a bit confused about the two previous methods. Could the authors give a short description of 'southern analysis' and 'bisulfite conversion followed by PCR' to highlight the drawbacks of these methods and how the new method (developed by the authors) improved the situation?
10) Lines 137-161: To me, the most of this part and Table 1 should be moved into the Materials & Methods section.
11) Lines 164-173: I was wondering of the effect of global Aza treatment on gene expression at other sides (see also my comment on the abstract section). Could the authors comment on this matter?
12) Lines 174-183: I suggest to clearly state which cell lines were used. It is mentioned in the Materials and Methods, but would certainly help to understand the manuscript better here as well.
13) Line 188: What is "this SunTag system"? Could be explained here (as the real description comes only later, it is not clear to a reader who is not familiar with the methodology).
14) Line 198: "significantly higher" - What does this mean? Which statistical test was used or do the authors just mean something like 'clearly higher, x-fold higher..' - I would just avoid to use the wording 'significant' without a statistical test, although that I am aware of that this wording is common in some fields.
15) Lines 312-319: Should be moved to the Discussion.
# Discussion
16) To me, on one hand, the Discussion section reads quite repetitive and contains many parts that were already mentioned in the Results section. On the other hand, the Results section contains parts that would better fit in the Discussion. I would recommend to streamline the writing and/or to include a paragraph with a clear conclusion to state the main take-home messages for this study.

Author Response
1. This has been added in the abstract line 16 and introduction line 80.
2. This has been modified accordingly.
3. This has been modified.
4.5.6.7. We take into account the reviewer' remarks and clarified all these points in the revised version.
8. The lengths of the bands of the ladder have been added in Figure 1C, Figure 2A, Figure 4C.

The abbreviations have been explained in Figure 1, caption.
9. Complementary information has been added in this part (lines 143-157).
10. The Table 1 and a part of this section has been moved into the Material and Methods section.
11. In this section control experiments have been performed with the global 5-aza treatment as a proof of concept of the method of analysis of the subbtelomeric DNA methylation. This treatment induced global genomic DNA demethylation. The analysis with this method of targeted demethylation on subtelomeric CpG island with the epigenetic tool is presented on figure 2A.
12. This has been specified (line 174).
13. We have introduced the SunTag system in this section (line 197-199).
14. We have deleted the word significantly, which was inappropriate.
15. This point has been addressed in the results section, as we think it is necessary to introduce in this section the supplementary data Figure 3.
16. According to the reviewer recommendation we have added at the end of the manuscript a short conclusion to resume the main take-home messages.